

# Maxent estimation of aquatic *Escherichia coli* stream impairment

Dennis Gilfillan[1], Timothy A. Joyner[2] and Phillip Scheuerman[1]

[1] Department of Environmental Health Sciences, East Tennessee State University, Johnson City, TN, United States of America

[2] Department of Geosciences, East Tennessee State University, Johnson City, TN, United States of America

## ABSTRACT

**Background**. The leading cause of surface water impairment in United States' rivers and streams is pathogen contamination. Although use of fecal indicators has reduced human health risk, current approaches to identify and reduce exposure can be improved. One important knowledge gap within exposure assessment is characterization of complex fate and transport processes of fecal pollution. Novel modeling processes can inform watershed decision-making to improve exposure assessment.

**Methods**. We used the ecological model, Maxent, and the fecal indicator bacterium *Escherichia coli* to identify environmental factors associated with surface water impairment. Samples were collected August, November, February, and May for 8 years on Sinking Creek in Northeast Tennessee and analyzed for 10 water quality parameters and *E. coli* concentrations. Univariate and multivariate models estimated probability of impairment given the water quality parameters. Model performance was assessed using area under the receiving operating characteristic (AUC) and prediction accuracy, defined as the model's ability to predict both true positives (impairment) and true negatives (compliance). Univariate models generated action values, or environmental thresholds, to indicate potential *E. coli* impairment based on a single parameter. Multivariate models predicted probability of impairment given a suite of environmental variables, and jack-knife sensitivity analysis removed unresponsive variables to elicit a set of the most responsive parameters.

**Results**. Water temperature univariate models performed best as indicated by AUC, but alkalinity models were the most accurate at correctly classifying impairment. Sensitivity analysis revealed that models were most sensitive to removal of specific conductance. Other sensitive variables included water temperature, dissolved oxygen, discharge, and $NO_3$. The removal of dissolved oxygen improved model performance based on testing AUC, justifying development of two optimized multivariate models; a 5-variable model including all sensitive parameters, and a 4-variable model that excluded dissolved oxygen.

**Discussion**. Results suggest that *E. coli* impairment in Sinking Creek is influenced by seasonality and agricultural run-off, stressing the need for multi-month sampling along a stream continuum. Although discharge was not predictive of *E. coli* impairment alone, its interactive effect stresses the importance of both flow dependent and independent processes associated with *E. coli* impairment. This research also highlights the interactions between nutrient and fecal pollution, a key consideration for watersheds with multiple synergistic impairments. Although one indicator cannot mimic the

Corresponding author
Dennis Gilfillan, gilfillan@etsu.edu

plethora of existing pathogens in water, incorporating modeling can fine tune an indicator's utility, providing information concerning fate, transport, and source of fecal pollution while prioritizing resources and increasing confidence in decision making.

# INTRODUCTION

Rapid urbanization of rural areas causes deterioration in water quality, rendering many water bodies unfit for their domestic and recreational use. An assortment of contaminants is introduced into aquatic systems, but pathogens represent the major cause of stream impairment in the United States (*United States Environmental Protection Agency, 2017*). Pathogens are difficult to measure directly because of their sporadic distribution, costly identification, and potential health risks to laboratory workers (*Field & Samadpour, 2007*). Most pathogens in aquatic systems stem from human and animal fecal wastes, including direct deposition of feces in water (*Vidon, Campbell & Gray, 2008*), run-off from land with fecal deposits (*Tyrrel & Quinton, 2003*; *Jamieson et al., 2004*), and sanitary sewer malfunctions (*Ferguson et al., 2003*; *McLellan & Eren, 2014*). To address the difficulties in monitoring specific pathogens, fecal indicator organisms (FIOs) are commonly used to assess the presence of fecal pathogens.

An effective fecal indicator is associated with the presence of specific pathogens, with a straightforward method for enumeration that correlates with magnitude and age of fecal pollution (*Savichtcheva & Okabe, 2006*; *Maier, Pepper & Gerba, 2009*). The use of FIOs such as fecal coliform bacteria and *Escherichia coli* are traditionally used for determining surface water pathogen impairment (*Yates, 2007*; *US Environmental Protection Agency, 2012*). Although these indicators assist in alerting populations when exposure to pathogens is likely, the current approach is limited by using a single indicator such as *E. coli* for a designated use (*Wade et al., 2003*; *Savichtcheva & Okabe, 2006*; *Field & Samadpour, 2007*). The cosmopolitan nature of *E. coli* in warm-blooded animals makes them impractical for source identification (*Field & Samadpour, 2007*; *Yates, 2007*; *McLellan & Eren, 2014*; *Blount, 2015*). The ability of *E. coli* to survive in soils (*Lasalde et al., 2005*; *Ishii et al., 2006*), algae (*Byappanahalli et al., 2003*), and sediments (*LaLiberte & Grimes, 1982*; *Alm & Burke, 2003*; *Drummond et al., 2015*) provide a reservoir for continued persistence and potential to naturalize (*Winfield & Groisman, 2003*; *Lasalde et al., 2005*; *Luo et al., 2011*). These characteristics and deficiencies emphasize the difficulty of single standard FIO monitoring for impairment, stressing the need for additional methods to evaluate source and mechanisms of FIO impairment.

In addition to the above issues, appropriately characterizing FIO impairment for regulation and decision-making is difficult due to complex fate and transport processes (*Benham et al., 2006*; *De Brauwere, Ouattara & Servais, 2014*; *Drummond et al., 2015*).

These complex fate and transport processes include transport through run-off and storm water (*Kistemann et al., 2002*; *Lipp et al., 2001*; *McKergow & Davies-Colley, 2010*), remobilization from sediments and hyporheic exchange (Drummond, 2015; *Dwivedi, Mohanty & Lesikar, 2016*), particle attachment (*Characklis et al., 2005*), and UV light exposure (*Sinton et al., 2002*). Additionally, ecological processes control FIO fate and transport through variable survival patterns of indicators and pathogens (*Anderson et al., 2005*; *Stott et al., 2011*), availability of nutrients and organic matter (*Surbeck, Jiang & Grant, 2010*; *Perkins et al., 2016*), and predation (*McCambridge & McMeekin, 1980*). Appropriately characterizing the physics and ecology driving fate and transport can better inform management decisions for total maximum daily load (TMDL) development, reduction of pollution, and allocation of resources.

Modeling provides flexible approaches to infer sources and processes associated with FIOs and other pathogens, overcoming some of the issues of the single indicator paradigm. Various statistical and machine learning models have been used to approach such problems of incorporating age of fecal pollution for source tracking or detection of viruses (*Brion, Neelakantan & Lingireddy, 2002*; *Black, Brion & Freitas, 2007*); identifying land use, environmental, and water quality parameters associated with FIOs and pathogens (*Brion & Lingireddy, 1999*; *Viau et al., 2011*; *Wilkes et al., 2011*; *Gonzalez et al., 2012*; *Gonzalez & Noble, 2014*; *Hall et al., 2014*; *Herrig et al., 2015*; *Lušić et al., 2017*); determining factors influencing particle attachment and virulence (*Piorkowski et al., 2013*); and optimizing microbial source tracking (*Belanche-Muñoz & Blanch, 2008*; *Ballestè et al., 2010*; *Smith, Sterba-Boatwright & Mott, 2010*; *Molina et al., 2014*). Some other applications of modeling include using turbidity or rainfall to predict *E. coli* concentrations at unmonitored sites (*Money, Carter & Serre, 2009*; *Coulliete et al., 2009*), estimating E. coli loads using physical, chemical, and biological factors within a neural network (*Dwivedi, Mohanty & Lesikar, 2013*), and hyporheic-groundwater interactions associated with transport of *E. coli* within sediments porewater (*Dwivedi, Mohanty & Lesikar, 2016*). Modeling can inform decision-makers concerning what drives impairment, addressing some of the shortcomings of a single indicator approach.

Maxent, a commonly used ecological niche model (*Phillips, Dudík & Schapire, 2004*; *Phillips & Dudík, 2008*), identified environmental variables associated with probability of *E. coli* stream impairment, making inferences concerning source and mechanisms driving fecal pollution. Although modeling *E. coli* using a machine learning model such as Maxent is not a novel approach, e.g., *Dwivedi, Mohanty & Lesikar, 2013*, this study is unique in the following ways: it focuses on how the water quality is associated with *E. coli* impairment in lower order streams, uses nonparametric bootstrapping as a probabilistic assessment of model performance based on the area under the curve (AUC) of the receiving operator characteristic (ROC), and uses loss of information as an indicator of sensitive variables. Ecological niche models have been utilized for species distribution (*Lozier, Aniello & Hickerson, 2009*), conservation of rare species (*Guisan et al., 2006*), invasive species (*Thuiller et al., 2005*), and disease vector epidemiology studies (*Boeckmann & Joyner, 2014*), but this is a new application of Maxent to microbial water quality. Additionally, developing models

in lower order streams has not been previously reported; this is important because water from low order streams is used for domestic water supply and recreation in many areas of the United States.

The motivation for using Maxent to predict *E. coli* impairment is to investigate how environment, i.e water quality parameters, shapes the niche of *E. coli* impairment based on a decision boundary; in this case, a water quality standard. A probabilistic procedure for univariate and multivariate model development is presented using nonparametric bootstrapping cross-validation. Univariate models generated action values, or environmental thresholds of impairment, to indicate potential *E. coli* impairment based on a single parameter. Multivariate models predicted probability of impairment given a suite of environmental variables, and jack-knife sensitivity analysis removed unresponsive variables in multivariate models to elicit a set of the most responsive water quality parameters. Using Maxent to model how water quality influences *E. coli* impairment aids in inferring source and mechanisms of fecal pollution. This approach allows for estimation of both linear and non-linear effects of water quality, demonstrates a probabilistic method for variable selection, and reframes the question from ''How much *E. coli* in our watershed?'' to ''what factors separate *E. coli* impairment from compliance?,'' which is useful when evaluating watershed decisions.

## METHODS

### Sampling sites and data collection

Sinking Creek is a 1st to 3rd Strahler order mixed-use stream that is noncompliant for state of Tennessee standards for fecal coliform and *E. coli* (*Tennessee Department of Environmental and Conservation, 2006*). Starting in August 2004, samples were collected by hand in August, November, February, and May of each year until August 2011 as a long-term monitoring plan at 14 sites in Sinking Creek, and samples were analyzed for 10 water quality parameters and populations of *E. coli* (Fig. 1).

Specific conductance (conductivity) and water temperature were measured using an Orion 115A+ conductivity meter (Thermo Fisher Scientific, Waltham, MA). The pH was measured using an EL2 portable pH meter (Mettler Toledo, Columbus, OH). Dissolved oxygen was collected using a YSI Model 55 dissolved oxygen meter (YSI Inc., Yellow Springs, OH). Samples for nitrates ($NO_3$), phosphates ($PO_4$), biochemical oxygen demand (BOD), alkalinity, and hardness were collected in clean 2 L polyethylene bottles and stored at 4 °C until laboratory analysis. A flow meter (Global Water, FP101) was placed in the center of the channel to measure stream velocity. Stream width was calculated where the stream velocity was measured, and depth was averaged over three points across the stream width. Velocity was multiplied by stream width and average depth to estimate discharge.

$NO_3$ and $PO_4$ analyses were performed in triplicate using colorimetric HACH$^{TM}$ methods (Hach, Loveland, CO) and reagents. $NO_3$ and $PO_4$ analyses were conducted by adding 10 mL of water to a vial containing NitraVer5 or PhosVer3 for the respective analyses. Vials were shaken to dissolve the reagent and samples were analyzed with a DR890 colorimeter (Hach, Loveland, CO) (*Hach Company, 2013*). Triplicate sample for alkalinity

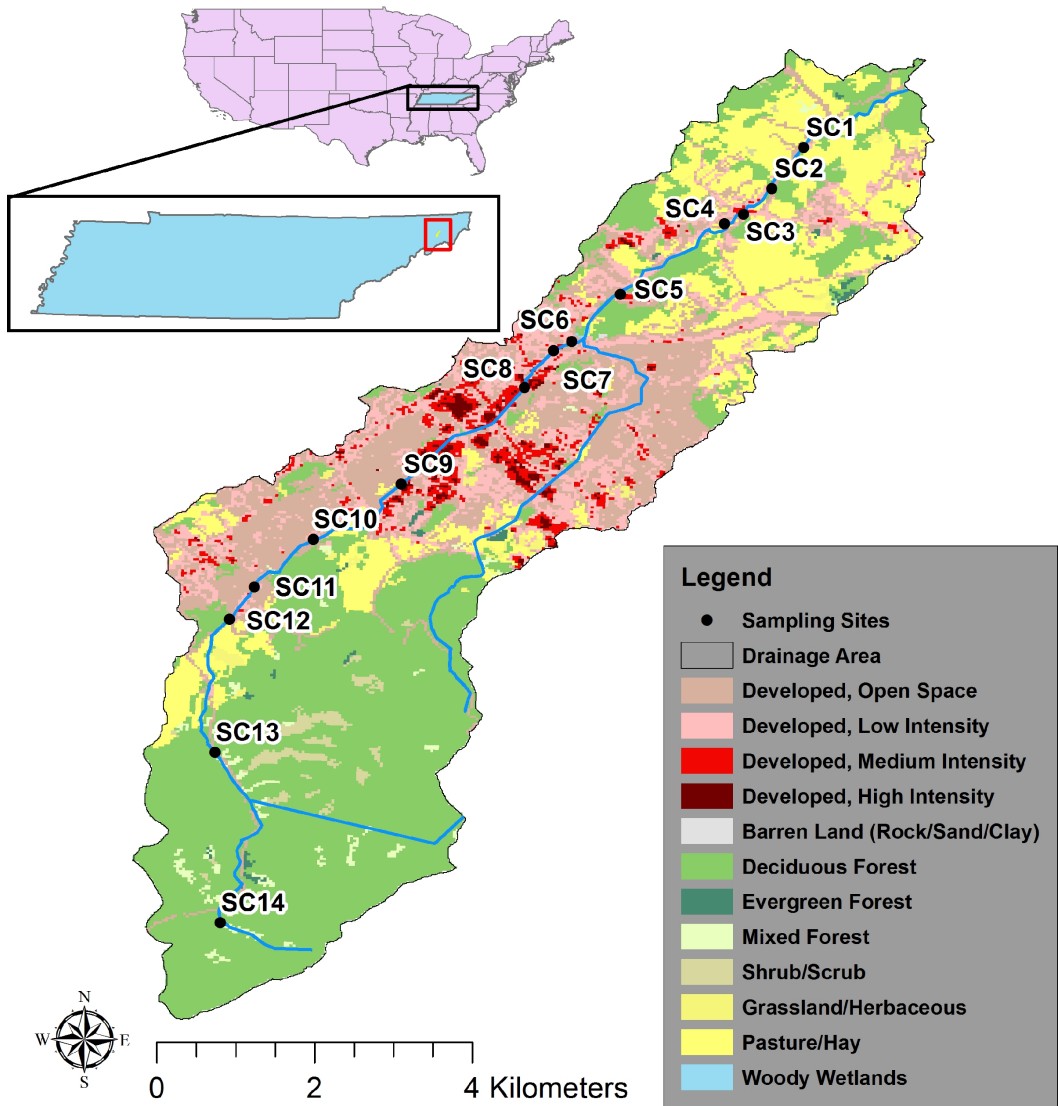

**Figure 1** **Map of sampling sites and watershed of the study area, Sinking Creek.** The inset map shows the United States and the state of Tennessee, and the location of Sinking Creek. Samples were taken from August 2004 to August 2011 during the months of August, November, February, and May. The outline represents the watershed boundary of Sinking Creek, and 2006 NLCD has been clipped to the watershed (*Fry et al., 2011*). Stream flows from its headwaters at SC14 downstream to SC1.

and hardness were determined using 100 mL sample volumes and a digital titrator (Hach, Loveland, CO) (*Hach Company, 2013*). Phenolphthalein and bromcresol green-methyl red indicators were used, and the sample was titrated with 1.6 N sulfuric acid to a grey-green endpoint (*Hach Company, 2006*). BOD was measured in triplicate using the 5-day BOD test (*American Public Health Association, 2005*). Populations of *E. coli* were determined using the Colilert defined substrate test. Briefly, 97 wells were filled with 100 mL of water sample with the Colilert substrate added. Samples were incubated for 24 hours, and wells that fluoresce under a UV light were considered positive for *E. coli,* and a most probable number

**Table 1  Sampling sites, land use, and *E. coli* concentrations in Sinking Creek.**  Percentage of each land cover types (Agricultural, Developed, and Forested) as well as *E. coli* Geometric means (GM), geometric standard deviations (GSD), and maximum and minimum values for each site used in the study.

| Sampling site | Agricultural land use (%) | Developed land use (%) | Forested land use (%) | E. coli GM (GSD) | Min, Max |
|---|---|---|---|---|---|
| SC1 | 15.6 | 36.4 | 47.3 | 254.5 (3.4) | 43.7,2398.8 |
| SC2 | 14 | 37.2 | 48.1 | 182.3 (6.1) | 17.4,39810.7 |
| SC3 | 9.7 | 38 | 51.5 | 137.1 (4.0) | 14.5,1737.8 |
| SC4 | 9.7 | 37.9 | 51.6 | 169.8 (5.7) | 8.5,23988.3 |
| SC5 | 8.7 | 38.1 | 52.4 | 140.0 (7.2) | 4.1,30903.0 |
| SC6 | 7.1 | 30.2 | 61.6 | 50.2 (8.3) | 0.5,8709.6 |
| SC7 | 7.1 | 30 | 61.8 | 36.7 (9.4) | 0.5,10232.9 |
| SC8 | 7.7 | 24.3 | 66.8 | 73.9 (5.3) | 10.7,8709.6 |
| SC9 | 7.4 | 19.9 | 71.4 | 110.3 (5.8) | 14.5,3981.1 |
| SC10 | 5.2 | 6.6 | 86.5 | 70.6 (5.2) | 6.2, 1995.3 |
| SC11 | 5.6 | 3.8 | 89 | 17.2 (9.9) | 0.5,1202.3 |
| SC12 | 5.8 | 2.1 | 90.3 | 91.3 (3.8) | 5.2,812.8 |
| SC13 | 0 | 1.1 | 96.5 | 7.8 (5.5) | 0.5,102.3 |
| SC14 | 0 | 0 | 100 | 5.0 (6.1) | 0.5, 245.5 |

estimate was made based on the number of positive wells in both the large and small wells (*American Public Health Association, 2005*). If a sample was in excess of the geometric mean United States recreational water quality criteria, the sample site was considered impaired. Impairment was based on recommendation 1, which is a threshold of 126 $\frac{CFU}{100mL}$ that corresponds to an illness rate of $\frac{36}{10,000}$ people (*US Environmental Protection Agency, 2012*).

To get an estimation of land use throughout the Sinking Creek watershed, land cover data were downloaded from the National Land Cover Dataset (NLCD) for 2006 (*Fry et al., 2011*). Each sampling site's drainage area was delineated using StreamStats version 3 from the United States Geologic Survey (*Ries III et al., 2017*). Land was grouped into 3 categories; forested, developed, and agricultural. Forested land includes the categories deciduous forest, evergreen forest, and mixed forest. Developed land use includes all developed categories; open space (less than 20% impervious surface), low intensity (20–49% impervious surface), medium intensity (50–79% impervious surface), and high intensity (80–100% impervious surface). Agricultural land included grassland/herbaceous and pasture/hay. The area of each land use was divided by the total area of the drainage area to get the percentage land use shown in Table 1, and sampling sites as well as land cover categories are shown in Fig. 1.

## Modeling background

Maxent is an iterative machine learning model commonly used for mapping species distributions (*Phillips, Dudík & Schapire, 2010*). Within the sample space, $x$, and given a set of environmental features (parameters), $f_1(x), f_2(x), \ldots, f_n(x)$, the Maxent distribution estimates a vector of feature weights, $\beta = (\beta_1 \beta_2, \ldots, \beta_n)$, that maximizes the entropy of the

raw distribution of impairments, $q_\beta(x)$, using a Gibbs distribution,

$$q_\beta(x) = \frac{\exp(\sum_{j=1}^{n} \beta_j f_j(x))}{Z_\beta} \tag{1}$$

where $Z_\beta$ is the normalization constant that ensures that $q_\beta(x)$ integrates to one over the study area (*Phillips, Dudík & Schapire, 2010*). This modeling approach is justified because it provides the maximum information concerning impairment. From a water quality management standpoint, this approach is beneficial because decision-makers and stake-holders are more concerned with factors associated with impairment rather than compliance when approaching fecal pollution monitoring and management.

Original features (parameters) can be transformed into quadratic, product, hinge, and threshold feature classes so that complex multivariate responses can be modelled (*Phillips & Dudík, 2008*), but Maxent incorporates L1 regularization to balance satisfying the constraints on the features while minimizing overfitting. L1 regularization is not unique to Maxent, and is used in many general linear models (*Elith et al., 2011*). A regularization parameter $\lambda_j$ smooths probability distributions, generating sparse solutions and removing unnecessary features; this shrinks weights to balance fit and complexity (*Elith et al., 2011*). Because of regularization, Maxent fits a penalized maximum likelihood model equivalent to minimizing the relative entropy dependent on the error-bound constraints

$$max_\beta \frac{1}{m} \sum_{i=1}^{m} \ln(q_\beta(x_i)) - \sum_{j=1}^{n} \lambda_j |\beta_j| \tag{2}$$

$$subject\ to \int q_\beta(x)\,dx = 1.$$

Where m is the number of positive samples, $n$ is the number of features, and $x$ is the feature vector for occurrence point $i$. Eq. (2) provides insight into how Maxent uses background data: the first term is larger for models that distinguish between impairment states the best. The second term represents the regularization, which gets larger as the weights $\beta_j$ increase, indicating a complex model more likely to over fit. The output of $q_\beta(x)$ is termed the raw distribution, but it is difficult to interpret due to its scale dependence. More background points result in smaller raw values because their sum cannot exceed 1 over a large amount of points (Phillips & Dudok, 2008; *Elith et al., 2011*). For this reason, the logistical output of the Maxent model, $P(x)$, will be used because it represents the probability of impairment given the sample space, $x$. This is a logistic model using the same set of weights $\beta$ with the intercept of the model determined by the entropy of $q_\beta(x)$, $H$. The model is shown in Eq. (3) below.

$$P(x) = \frac{e^H q_\beta(x)}{1 + e^H q_\beta(x)}. \tag{3}$$

### Univariate models

Data were processed using a list-wise deletion process, where individual samples from a site were removed if they were missing a parameter measurement due to laboratory errors, equipment malfunctions, calibration issues, or sites being dry at the time of sampling. A

sample of 100 bootstrapped models were developed, and 20% was subsampled for testing validation. Bootstrapping is a nonparametric resampling technique to make inferences about a population based on resampling from a set population, generating population level statistics, while providing an estimate of uncertainty of those statistics (*Campolongo, 1997*). For this modeling approach, all background points are used in the development of the null model, and the impaired samples are bootstrapped. Although Maxent can incorporate a wide variety of feature classes, only linear and quadratic feature classes were used to develop action values, or thresholds of impairment. The rationale for using these types of feature classes is for ease of generating action values as well as to assess both linear and non-linear effects of single parameters.

The AUC was calculated for the training and testing datasets. The AUC is a metric of performance for binary classification. the true positive prediction rate (sensitivity) and false positive prediction rates (1–specificity) of each sample are plotted as a ROC for different decision boundaries, and the area under that ROC is integrated. An AUC of 0.5 indicates that the model is no better than random chance, and a value of 1.0 indicates perfect model performance (*Zweig & Campbell, 1993*; *Zou, O'Malley & Mauri, 2007*).

The decision boundary (logistic threshold) between impaired and unimpaired samples should maximize accurately predicting impairment (exceedance of the *E. coli* criteria) while balancing correct negative predictions (*Bean, Stafford & Brashares, 2012*). Therefore, maximum test sensitivity and specificity was defined as the appropriate decision boundary. A low sensitivity would indicate poor performance in identifying impairment, while a low specificity would indicate an overcautious model in which resources might be wasted in remediation of false positives. Accuracy for Maxent models was calculated as follows: $\frac{TP+TN}{TP+TN+FP+FN}$ , where TP are true positives, TN are true negatives, FP are false positives and FN are false negatives. Significance of the univariate model was determined by calculating the $\chi^2$ statistic for each confusion matrix, with the null hypothesis being that the classifier was no better than random chance.

Action values (environmental thresholds) are conditions in which a parameter (variable) is at the threshold of impairment, indicating potential exceedance of the *E. coli* standard. Action values were calculated for significant ($p < 0.01$) univariate models by averaging bootstrapped weights and estimating the parameter value at which the probability of impairment equals the logistic threshold. Figure 2 demonstrates the concepts of the AUC performance metric, selected decision boundary, and the concept of the action value in relation to the selected decision boundary and Maxent model function (Eq. (3)).

### Multivariate models and sensitivity analysis

Although some authors state that collinearity is not as problematic in Maxent compared to traditional regression approaches, collinearity was explored and subsequently removed using Pearson correlation coefficients (*Elith et al., 2011*). Variables that were highly correlated ($r > 0.8$) were evaluated to determine which variable to include based on expertise, connections to previous models, and accuracy metrics within the analysis. The initial multivariate model included all noncollinear variables and were developed using 100 bootstrapped samples like the univariate models, with the addition of product feature

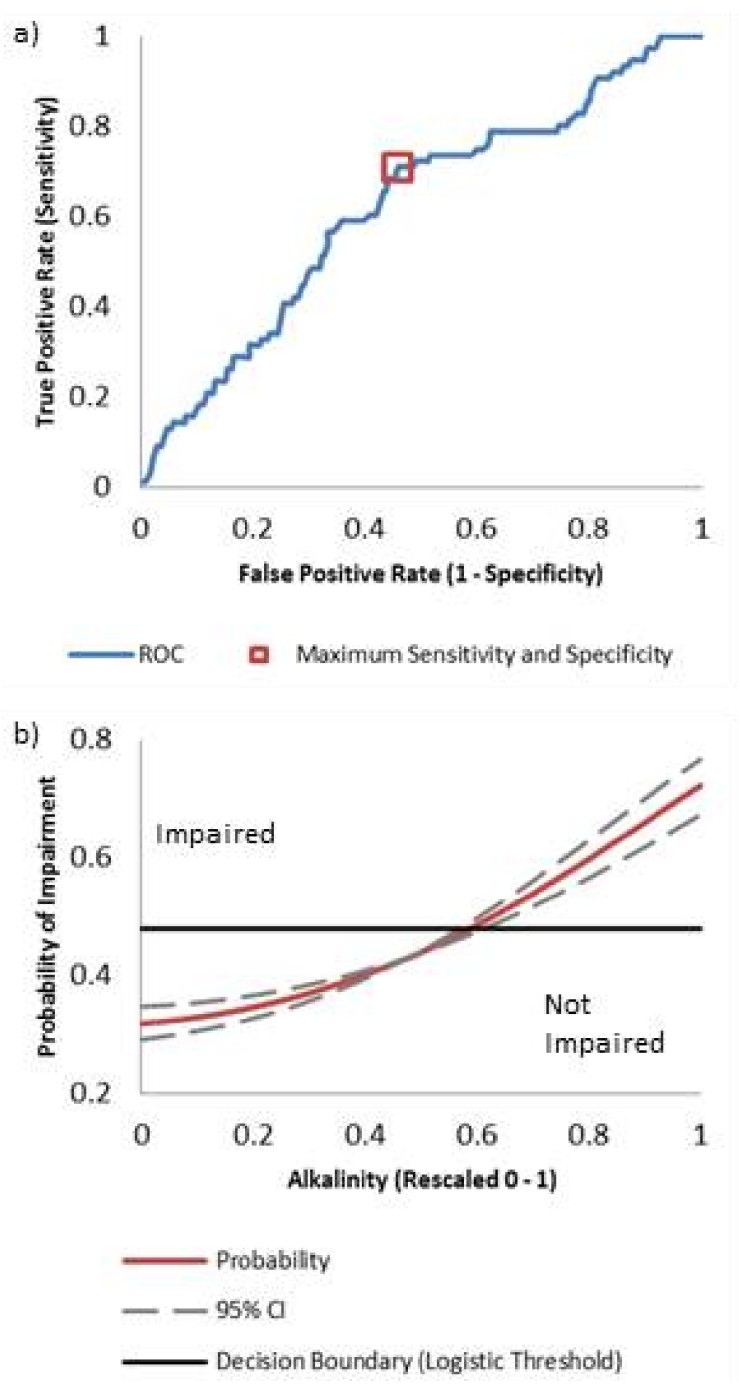

**Figure 2  Theoretical plots to illustrate the concept of the ROC, decision boundaries, and action values.** (A) Plot of an ROC curve, where the $x$-axis represents the false positive rate, or the compliment of the specificity, and the $y$-axis represents the true positive rate, the sensitivity. The curve is integrated to obtain the AUC, the performance metric for each of the models. The box represents the point at the decision boundary (B) Theoretical plot of a univariate Maxent model function (Eq. (3)) with values for alkalinity rescaled from 0 to 1. The solid red line represents Eq. (3), the dotted lines represent the upper and lower 95% confidence intervals, and the horizontal black line represents the decision boundary. The action values, or environmental thresholds, and associated confidence intervals are the intersections between the results of Eq. (3) and the decision boundary.

classes to incorporate variable interaction. Average variable contribution was determined by calculating the increase in information gain associated with a change in each feature for each iteration of the model algorithm, normalized to percentages. The permutation importance of a feature is an indicator of variable sensitivity. In each model run, the feature training presence and background data are randomly permutated, and the resulting drop in training AUC is normalized to percentages for each variable and averaged over the 100 bootstrapped runs.

A jack-knife sensitivity analysis was used to determine the best subset of covariates to include in a trimmed model. Each variable was removed from the analysis, and a comparison was made to determine if the removal of a variable caused a significant ($p < .05$) change in training or testing information gain. Student's $t$-tests were performed on each jack-knifed model to evaluate significance, and variables were included if the information gain from either the testing or training sets decreased; decrease in information would correspond to a significant loss of information, providing criterium for inclusion of the variable in final models.

## RESULTS

### Univariate model performance

The sampling program resulted in 29 sampling trips over 14 sites, allowing for a potential of 406 samples for analysis. 127 samples were removed due to missing information in the dataset, leaving 279 total samples for model development. This included 95 impairments, identified by exceedance of the *E. coli* recreational water quality standard. Table 1 presents the summary statistics for *E. coli* and the associated land use in each sampling site's drainage area. Each training set included 279 background points, 76 points for training, and 19 points to evaluate performance on testing data.

Table 2 summarizes the training and testing performance of the univariate Maxent models used to identify *E. coli* impairment based on environmental variables. Water temperature performed best based on AUCs, but had lower accuracy than conductivity, dissolved oxygen, and alkalinity. The plausible explanation of these differences is the latter variables had higher specificity rather than sensitivity at the chosen decision boundary (Table S2). Accuracy was found to be highest for alkalinity and lowest for pH.

Action values were developed for 8 significant univariate models by solving for the value of the variable when probability of impairment equals the logistic threshold. For example, the action value for alkalinity is 128 mg/L. This means that *E. coli* impairment is likely to occur when alkalinity is observed to be higher than this threshold. Action values and 95% confidence intervals are included in Table 2. Action function graphs for each significant univariate model are presented in Fig. S1 to aid in interpretation of Table 2, and summary statistics for each variable are given in Table S1.

### Multivariate model performance

Pearson correlations ranged from −0.269 to 0.834, with three variables identified as collinear; alkalinity, conductivity, and hardness. Conductivity was selected because of its use in previously developed fecal indicator models (*Wilkes et al., 2011*; *Gonzalez et*

**Table 2  Summary of training and testing performance of Maxent models based AUC metrics, accuracy based on maximum test sensitivity and specificity decision boundary (logistic threshold), and action values with 95% confidence intervals.** If an upper bound of a confidence interval exceeds the maximum sampling value for a set of data, the maximum value is given.

| Variables | Training AUC (± SE) | Testing AUC (±SE) | Accuracy | Action values (x) ¥ (95% CI) |
|---|---|---|---|---|
| Alkalinity ($\frac{mg}{L}$) | 0.616 (0.003) | 0.620 (0.006) | 68.5 | $x > 129\frac{mg}{L}$ (125, 134) |
| BOD ($\frac{mg}{L}$) | 0.572 (0.004) | 0.554 (0.008) | 60.6 | $x < 0.976\frac{mg}{L}$ (0.825, 1.09) $x > 6.19\frac{mg}{L}$ (4.51, 6.43) |
| Conductivity (µS) | 0.628 (0.003) | 0.638 (0.006) | 65.6 | $x > 306$ µS (287, 315) |
| Dissolved Oxygen ($\frac{mg}{L}$) | 0.635 (0.003) | 0.640 (0.007) | 67.7 | $x < 9.39\frac{mg}{L}$ (8.68, 10.6) |
| Discharge ($\frac{m^3}{s}$) | 0.556 (0.004) | 0.553 (0.006) | 63.8 | * |
| Hardness ($\frac{mg}{L}$) | 0.632 (0.003) | 0.627 (0.006) | 59.9 | $x > 132\frac{mg}{L}$ (122, 152) |
| NO$_3$ ($\frac{mg}{L}$) | 0.581 (0.004) | 0.579 (0.007) | 63.4 | $x > 1.78\frac{mg}{L}$ (1.63, 1.84) |
| pH | 0.571 (0.003) | 0.562 (0.006) | 55.6 | * |
| PO$_4$ ($\frac{mg}{L}$) | 0.581 (0.004) | 0.580 (0.008) | 63.8 | $0.0642\frac{mg}{L} < x < 7.80\frac{mg}{L}$ (0.0873, 0.766) ˇ (6.27, 9.01) ˆ |
| Water Temperature (°C) | 0.666 (0.003) | 0.670 (0.005) | 65.2 | $x > 12.4$ °C (11.3, 15.5<x<20.0) |
| 8-variable model | 0.770 (0.002) | 0.709 (0.005) | 78.5 | |
| 5-variable model | 0.753 (0.002) | 0.723 (0.006) | 77.8 | |
| 4 variable model | 0.750 (0.002) | 0.726 (0.005) | 77.8 | |

**Notes.**
\*Model was not significant.
[a]Values of the variables that corresponded to impairment.
[b]95% CI for the lower bound of the action value.
[c]95% CI for the upper bound of the action value.

*al., 2012*; *Gonzalez & Noble, 2014*; *Piorkowski et al., 2013*). The 8-variable model displayed improved accuracy on all univariate models. Variable contribution was dominated by water temperature, conductivity, and discharge in the 8-variable model, with water temperature contributing 36.4% of the information, and conductivity and discharge accounting for 22.6% and 12.1% of the information, respectively. The permutation importance for the 8-variable model demonstrated a similar pattern. A summary of accuracy metrics is shown in Tables 2 and 3 illustrates the contribution of each variable in the multivariate models.

Conductivity was the most sensitive parameter based on sensitivity analysis, with other sensitive parameters including water temperature, dissolved oxygen, discharge, and NO$_3$. The removal of dissolved oxygen improved model performance based on testing AUC, justifying development of two optimized multivariate models; a 5-variable model including all sensitive parameters, and a 4-variable model that excluded dissolved oxygen.
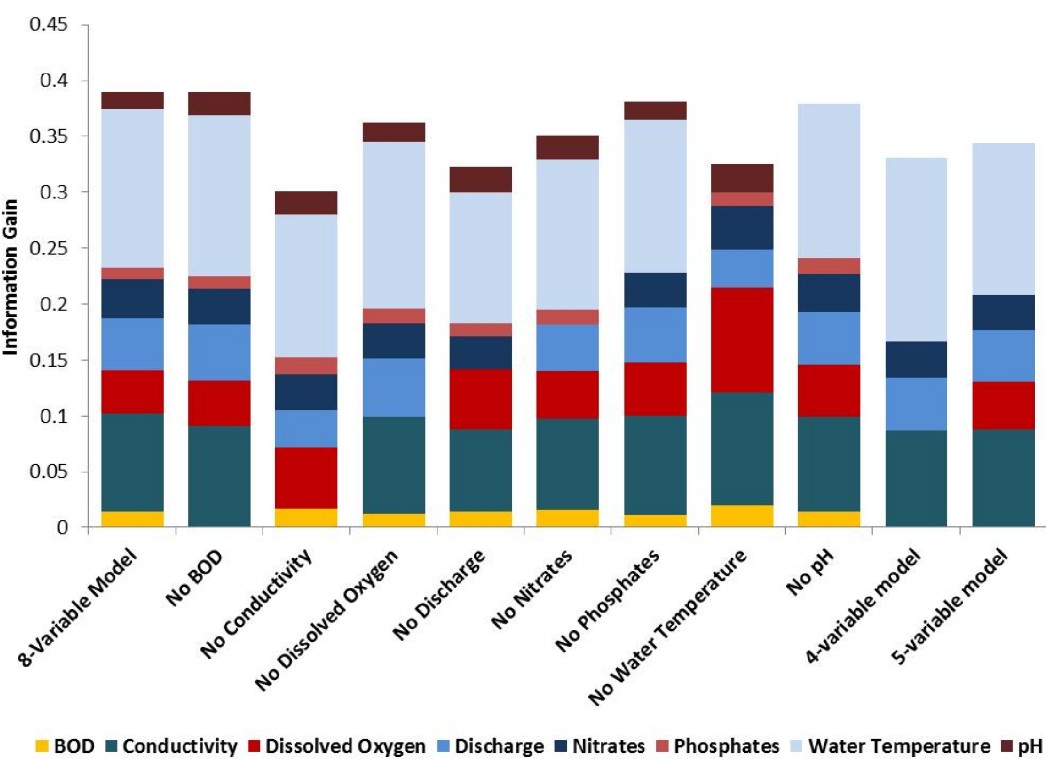

**Figure 3  Bar graph displaying results of jack-knife sensitivity analysis.** Each color represents the information gain contributed for each parameter in the model, and features are removed one at a time to assess their importance in the trimmed model.

**Table 3  Variable contribution and permutation importance for the multivariate models, normalized to percentages.**

| Variable | 4-variable model | | 5-variable model | | 8-variable model | |
|---|---|---|---|---|---|---|
| | Percent contribution | Permutation importance | Percent contribution | Permutation importance | Percent contribution | Permutation importance |
| BOD | | | | | 3.6 | 5.9 |
| Conductivity | 26.2 | 23.0 | 22.6 | 22.3 | 25.6 | 27.5 |
| Discharge | 14.5 | 22.0 | 12.1 | 20.1 | 13.4 | 21.6 |
| Dissolved Oxygen | | | 9.9 | 5.2 | 12.3 | 6.6 |
| $NO_3$ | 9.5 | 8.5 | 8.9 | 8.6 | 8.9 | 10.3 |
| pH | | | | | 3.9 | 1.7 |
| $PO_4$ | | | | | 2.7 | 2.5 |
| Water temperature | 49.9 | 46.5 | 36.4 | 33.7 | 39.7 | 34.0 |

Accuracy of the 5- and 4- variable optimized model was 77.8%. The patterns of variable contribution were consistent in each model, with water temperature accounting for most of the information gain in each model. Figure 3 shows the variable contribution for the initial multivariate models, each model run during the sensitivity analysis, and the final 4-variable and 5-variable models produced. The information gain for each model is also shown within this figure.

Response surfaces were developed for each of the model runs to assess spatiotemporal trends. Each grid within the surface represents a single sample, with each sampling site representing a single column. The columns are oriented in a downstream fashion, with headwaters sites starting on the left (SC14) and sites further downstream existing on the right (SC1). The temporal scale is represented by the rows, with each row indicating a specific sampling trip. Although the data resolution is coarse, the goal is to demonstrate the potential of visualizing trends in the probability of impairment over space and time. Figure 4 displays the response surface for the estimated probability of impairment for the 4-variable model and the 5- and 8-variable models are shown in Fig. S2. Classification performance for the univariate models and multivariate models is shown in Table S2. Mean probabilities for the 8-, 5-, and 4-variable model were 0.338 (95% CI: 0.319, 0.358), 0.353 (0.334, 0.373), and 0.359 (0.340, 0.378). Generally, the sites influenced by the greatest amount of developed or agricultural land use (SC5–SC1) had the highest probability of impairment. August had the highest probability of impairment, followed by May, November, and February. Mean probability of impairment and associated 95% confidence intervals are shown in Table S3.

## DISCUSSION

Over 170,000 miles of US rivers and streams are listed as pathogen impaired based on FIOs. To address these impairments, characterization of sources and transport mechanisms is necessary (*United States Environmental Protection Agency, 2017*), and statistical models can be used as an inferential tool to overcome these issues. We applied Maxent to identify individual and interacting factors influencing *E. coli* fate and transport that resulted in impairments using univariate and multivariate approaches. In this particular stream, water temperature, conductivity, discharge, and $NO_3$ were found to be the most influential group of factors driving fecal pollution. The results indicate that seasonality and agricultural run-off are the suggested causes of impairment in this watershed. Seasonality is demonstrated by influence of temperature in the models, whereas the influence of agricultural run-off is suggested by the other variables and the association between land use and *E. coli* in the watershed. Even small increases in agricultural land cause substantial increases in *E. coli* concentrations (Table 1), whereas similar increases in developed land do not have the same pronounced effect. This study highlights the need for multi-month sampling across a stream continuum to truly estimate spatiotemporal variability associated with impairment.

The fact that water temperature dominated the information in this model suggests that seasonality plays an important role in *E. coli* survival. Although fecal indicators and pathogens have been found to possess diverse temperature-survival relationships (*Hofstra, 2011*; *Sterk et al., 2013*), the high August probability for *E. coli* impairment indicates favorable conditions for long-term survival in the summer. Warming due to climate change could exacerbate this condition by increasing those favorable conditions (*Weniger et al., 1983*; *Atherholt et al., 1998*; *Patz et al., 2000*; *Guzman Herrador et al., 2015*). However, August was not the only month with numerous *E. coli* impairments. Therefore, monitoring for FIOs only in the summer months could distort estimates of impairment in watersheds with year-round users.

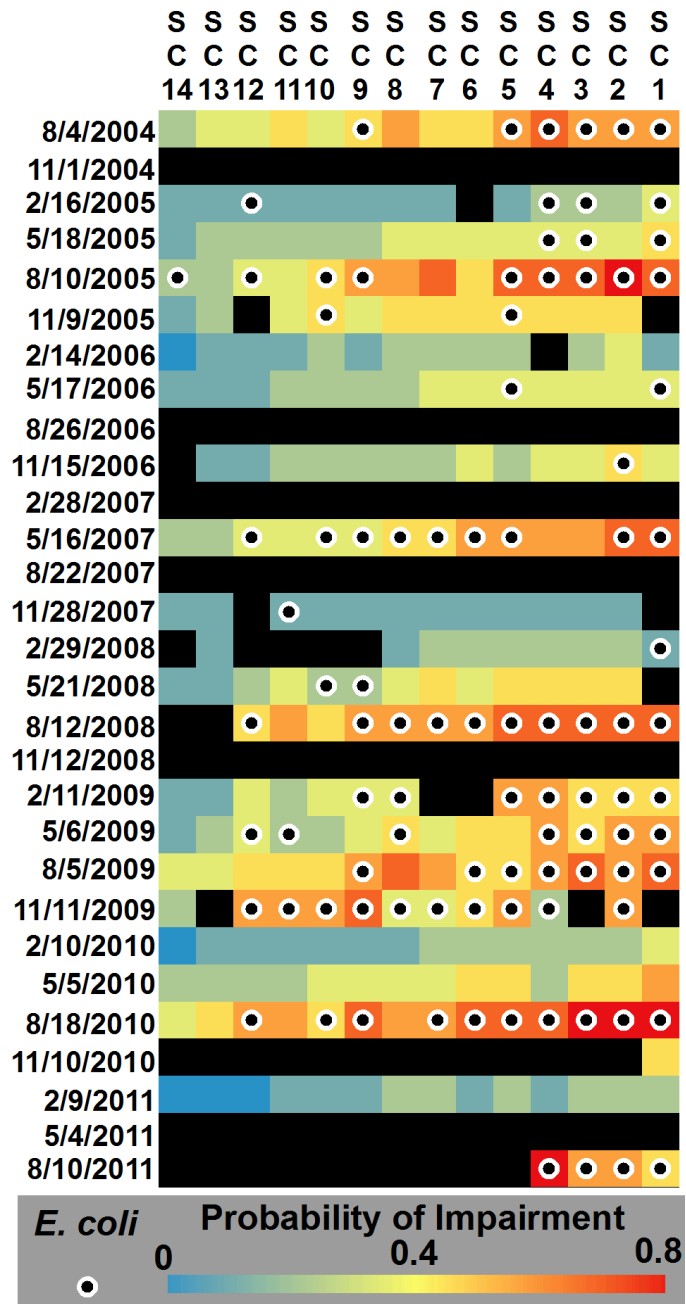

**Figure 4 Response surface for the 4-variable Maxent model.** Surface shows the probability of impairment for each sample for the monitoring program. This represents the mean probability of 100 bootstrapped runs. Rows are oriented by each sampling period, while columns represent each sampling site over the length of the stream; left to right indicates flow direction. Black cells denote samples in which a parameter was missing and were excluded from analysis, while circles with black centers represent samples in which a stream would be identified as impaired in the study.

Although discharge was not predictive of *E. coli* impairment alone, its interactive effect stresses the importance of both flow dependent and independent processes associated with *E. coli* impairment. Dissolved solutes such as $NO_3$ and ions measured through conductivity are largely discharge-dependent; however, FIOs are not as strongly dependent on discharge. This flow independence is due to additional ecological mechanisms such as nutrient limitation and competition (*Surbeck et al., 2006*; *Drummond et al., 2015*). Various forms of nitrogen are associated with increased concentration of FIOs in certain environments (*Carrillo, Estrada & Hazen, 1985*; *Herrig et al., 2015*), and results of the Maxent models suggest that nutrient loading in the form of $NO_3$ contributes to *E. coli* impairment in Sinking Creek. Other studies have found that dissolved organic carbon can affect magnitude and extent of fecal indicators (*Surbeck, Jiang & Grant, 2010*; *Blazewicz et al., 2013*; *Cloutier, Alm & McLellan, 2015*), but this was not collected during this sampling program and was found to be insignificant using BOD as a surrogate for organic pollution. This interaction between nutrient levels and fecal pollution highlights the potential for synergistic effects of different sources of pollution, suggesting a limitation of TMDL development when only considering one pollutant at a time.

Although machine learning application to microbial water quality problems is not unique, this study presents some beneficial techniques in this area of research. First, it demonstrates the ability to open the black box of Maxent, using action values to predict threshold of impairment based on a single variable. Multivariate action functions can be developed as well, but is not presented in this manuscript. The probabilistic approach to model validation and variable selection allows for inclusion of uncertainty, improving on deterministic methods traditionally used for validation and criteria for variable inclusion. Probabilistic methods have been used in TMDLs (*Borsuk, Stow & Reckhow, 2002*), frequency of water quality posting errors (*Kim & Grant, 2004*), and uncertainty of different fecal indicator methodologies (*Gronewold et al., 2008*); this paper adds to this framework through identifying the probability of stream impairment given a set of environmental variables. This improves confidence in decision-making for implementation of monitoring, management, and remediation strategies. Modeling microbial water quality is a challenge no matter the method used, but this study demonstrates that Maxent provides a valid approach to understand the factors driving impairment.

Streams are dynamic systems with multiple flow regimes, confounding an already difficult modeling process. Understanding how models behave in extreme situations is useful for regulation, monitoring, and management of these ecosystems. Over the long-term study periods, samples from both drought and high water conditions were captured. Maxent has been suggested as a strong prediction of extreme values (*Petrov, Guedes Soares & Gotovac, 2013*), and this study found that Maxent sufficiently predicted impairment during the high flow sampling date of November 11, 2009. Depending on which multivariate model was used, accuracy ranged from 72.8% to 90.9% for this sampling date. Five sampling dates resulted in at least one site being dry, indicating drought-like conditions. Maxent correctly predicted impairment in these situations 62.2% to 73.0% of the time. This suggests that Maxent can be useful for certain extreme situations, but is highly dependent on the environmental variables used for prediction.

While this study presents proof of concept of using Maxent to infer source and mechanisms of impairment, there are some limitations to this study. Although the dataset has a large time scale (8 years), only collecting from 4 months makes the resolution coarse, reducing the scale at which inferences can be made. The list-wise deletion of samples before univariate modeling removed some data that could inform each of those models; however, using the same series of data in the multivariate models and list-wise deletion are commonly used procedures in statistical models. Future applications of Maxent will improve on the coarse resolution of the data by using monthly and potentially weekly sampling approaches, and research will be developed as to the best approach for handling missing data in Maxent. While AUC scores above 0.70 indicate good model fit, only considering physiochemical water quality parameters limits the potential to accurately predict impairment; however, this study demonstrates that these parameters are informative as a proof of concept for using Maxent as a modeling approach. Future areas of research include using Maxent to optimize water quality monitoring to identify causes of impairment with FIOs and specific pathogens in the most cost-effective way using a variety of microbial, chemical, and physical parameters.

It is a difficult task to develop and implement remediation strategies in watersheds with many diffuse causes of fecal impairment, but modeling can increase confidence in decision making through inferring mechanisms and sources of fecal pollution. Incorporating environmental variables into models allows for insights into the ecology of fecal indicators, identifying causes of chronic FIO impairment. Although one indicator cannot mimic the plethora of existing pathogens in water, incorporating modeling can fine tune an indicator's utility, ultimately informing the public concerning health risks, and aiding in overcoming the shortcomings of a single indicator monitoring strategy.

## CONCLUSIONS

Characterizing *E. coli* impairment is essential because of the plethora of streams polluted with fecal wastes. This study used Maxent to identify water quality parameters associated with *E. coli* impairment in a low-order, mixed-use watershed. Univariate models generated action values, or thresholds of impairment, based on single parameters, while multivariate models extracted information concerning multivariate interaction. We presented a probabilistic approach to sensitivity analysis, improving confidence in variable selection. Maxent presents a flexible machine learning approach to aid in understanding mechanisms and sources of fecal pollution as well as a host of other complex decision boundary problems. We demonstrated that:

- Models using alkalinity and water temperature were found to be either the most accurate or best performing univariate models; this stresses the importance of discharge composition and seasonality in *E. coli* impairment. Discharge, however, was not an influential univariate parameters by itself, stressing the importance of flow-independent processes that correlate with impairment.

- Sensitivity analysis indicated that the most information was lost when conductivity was removed from the multivariate models, and water temperature, discharge, dissolved oxygen, and $NO_3$ represent other sensitive parameters sensitive to *E. coli* impairment in this watershed.
- Results suggest that *E. coli* impairment in this stream is driven by seasonality and agricultural run-off. This suggests that multi-month sampling along a stream continuum is essential to characterize spatiotemporal variability, importance of flow in relation to other water quality parameters, and the potential synergistic effect of nutrient and fecal pollution.
- Incorporating modeling can fine tune an indicator's utility, informing the public concerning human health risks, enhancing our understanding of FIOs, assisting in water quality decision-making, and providing input variables for quantitative microbial risk assessment.

### Abbreviations

| | |
|---|---|
| **AUC** | Area under the curve |
| **BOD** | Biochemical Oxygen Demand |
| **FIO** | Fecal Indicator Organism |
| **NLCD** | National Land Cover Dataset |
| **ROC** | Receiver Operating Characteristic |
| **TMDL** | Total maximum daily load |

## ACKNOWLEDGEMENTS

The authors thank Brian Evanshen for oversight of sample collection and data management.

### Funding

This work was supported by a contract with the Tennessee Valley Authority (Award # 0025252) and a grant from the East Tennessee State University's school of Graduate Studies and Graduate Council. The funders had no role in study design, data collection and analysis, decision to publish, or preparation of the manuscript.

### Grant Disclosures

The following grant information was disclosed by the authors:
Tennessee Valley Authority: # 0025252.
East Tennessee State University's school of Graduate Studies and Graduate Council.

### Competing Interests

The authors declare there are no competing interests.

## Author Contributions

- Dennis Gilfillan and Timothy A. Joyner conceived and designed the experiments, performed the experiments, analyzed the data, contributed reagents/materials/analysis tools, prepared figures and/or tables, authored or reviewed drafts of the paper, approved the final draft.
- Phillip Scheuerman authored or reviewed drafts of the paper, approved the final draft, allowed access to data, mentored, etc.

## Data Availability

The raw data are provided in the Supplemental Files.

## Supplemental Information

Supplemental information for this article can be found online at http://dx.doi.org/10.7717/peerj.5610#supplemental-information.

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
