# Peer review of "Maxent estimation of aquatic Escherichia coli stream impairment"

_PeerJ, doi:10.7717/peerj.5610_

## Round 0.1 · original submission · Major Revisions

The reviewers both appreciated the contribution of your paper but strongly emphasized the need for additional clarity in motivation, background, and discussion of results. Please revise the document according to these suggestions to strengthen the manuscript.

Reviewer 1 ·

Basic reporting

The paper is interesting but it suffers from a lack of clarity. Even in the abstract I find it hard to identify the main purpose and finding of the research. I recommend using Active voice rather than Passive.

For example in the abstract the authors write: Univariate models showed Water temperature as the best performing individual predictor, but models showed lower accuracy than conductivity, dissolved oxygen, or alkalinity. Accuracy was found to
be highest for alkalinity, and was found to be lowest for pH. - Does this mean accuracy of the models was highest for alkalinity? Is accuracy important? If not, don't mention it. If it is then don't mention the other variables because you're loosing your message. I encourage you to use clear statements. What are environmental controls? Are these modifiable factors implicated in stream impairment? Is temperature an environmental control? can you control that?

Experimental design

The methods and results are sufficient but again they suffer from a lack of clarity and an abundance of jargon. MaxEnt has been described in detail elsewhere. It wasn't even entirely clear why you were using MaxEnt if you have data from all these sites, are trying to predict outside of these sites or find a relationship between the variables and the outcome?

A clear statement such as (not sure if this even the case) We used MaxEnt to identify the probability of stream impairment given a set of environmental variables would go along way to improving readability.

Validity of the findings

It is difficult to interpret figure 4, what are the black and white circles and why are some of the grids black?

You have a testing AUC around .70 which is pretty for machine learning algorithms using environmental data low but don't discuss the implications of this in the context of policy.

Reviewer 2 ·

Basic reporting

This is a useful paper that demonstrates the opportunities in site-specific data analysis of E. coli concentrations in streams. Should such analysis be repeated for other sites, substantial progress could be achieved in understanding and mitigating surface water impairment. However, the manuscript has weaknesses in that the presentation is very general, and therefore targets a limited audience with the previous background in machine learning techniques. Moreover, the manuscript is also lacking any insightful discussion. For example, why different models (e.g., four variables vs. eight variables) do differently? Is that because of different capabilities of models or some physical understanding? Lastly, it is not clear how these models do in extreme situations. This might be useful for regulatory bodies and recreational activities. I also suggest bringing forth novelty in this manuscript as most of the points shown in the manuscript are well established (see some articles given below). These limitations preclude publication of this manuscript.

Experimental design

Experimental design is appropriate, but some more details are required as indicated in the attached PDF file.

Validity of the findings

Findings are valid, but most of the findings are well known. So authors need to bring forth the novelty of their work.

Additional comments

Additional comments are as follows:
1. Your introduction needs more detail. I suggest that you improve the description at lines 85- 98 to provide more justification for your study. Specifically, authors should include previous studies (Vidon et al. 2008, McKergow and Davies-Colley 2009, Money et al. 2009; Dwivedi et al. (2013) - just to name a few recent papers). There is a rich body of literature that this manuscript does not acknowledge or incorporate into the discussion, which causes the reader to question the diligence of the authors.

• McKergow, L. A., and R. J. Davies-Colley. 2009. Stormflow dynamics and loads of Escherichia coli in a large mixed land use catchment. Hydrological Processes:doi: 10.1002/hyp.7480.

• Money, E. S., G. P. Carter, and M. L. Serre. 2009. Modern Space/Time Geostatistics Using River Distances: Data Integration of Turbidity and E. coli Measurements to Assess Fecal Contamination Along the Raritan River in New Jersey. Environmental Science & Technology 43:3736-3742. doi:10.1021/es803236j.

• D Dwivedi, BP Mohanty, BJ Lesikar, Estimating Escherichia coli loads in streams based on various physical, chemical, and biological factors - Water resources research, 2013

• D Dwivedi, BP Mohanty, BJ Lesikar, Impact of the linked surface water-soil water-groundwater system on transport of E. coli in the subsurface - Water, Air, & Soil Pollution, 2016


2. L100 – provide reference of the Maxent model.
3. L 100-106 – it does not sound like a new approach, but Dwivedi et al. (2013) used water quality data to model E. coli in streams using a machine learning approach. Here, authors are just using a different machine learning technique. Authors need to strengthen the rationale why to use a different model here.
4. L115 – the goal of the present work should not to show whether a model works or not. The goal should be to fill knowledge gaps and connect to a bigger picture.
5. L125-171 Sampling sites and data collection – provide more details such as the number of samples, the frequency for each water quality parameter. Maybe, a table describing these details would be a better choice.
6. L177—why do you maximize the entropy of the raw distribution of impairments to get the optimum weights? Provide justification for that in a natural system like yours.
7. L211 – Authors need to provide details about which samples were deleted. If deleted samples were associated with extreme values, then how good your bootstrap samples are? This would lower the usefulness of the model. Maybe, authors could include synthetic data to analyze extreme cases to show the validity of their model across the spectrum.

---

## Round 0.2 · accepted · Accept

Although I agree with the reviewer's comment that it is important to ascribe physical understanding to the results of a probabilistic analysis where possible (to avoid treating ML methods as purely black boxes), in my opinion the current manuscript does do so at a basic level. In particular, in the first paragraph of the discussion the authors discuss the implications of the variables that were identified as most influential, specifically that seasonality and land use (agricultural runoff) are primary causes of impairment. Accordingly, and as the revised version addresses well the comments of both original reviewers, I recommend acceptance of the current version.

# Reviewer 2 ·

Basic reporting

Authors have done an excellent revision.

Experimental design

Same

Validity of the findings

-- same --

Additional comments

Authors have done an excellent job of revising the paper. I agree that parsimonious models are preferable. However, I am not convinced that it is not possible to provide a discussion on model performance differences based on the selection of variables. As such the artificial intelligence (AI) based approaches are nothing but black boxes, so efforts should be made to enhance process understanding.

External reviews were received for this submission. These reviews were used by the Editor when they made their decision, and can be downloaded below.